# Recognising Sepsis as a Health Priority in Sub-Saharan African Country: Learning Lessons from Engagement with Gabon’s Health Policy Stakeholders

**DOI:** 10.3390/healthcare10050877

**Published:** 2022-05-10

**Authors:** Bayode Romeo Adegbite, Paul Kawale, Levi Kalitsilo, Shevin T. Jacob, Jamie Rylance, Ayola Akim Adegnika, Martin Peter Grobusch

**Affiliations:** 1Centre de Recherches Médicales de Lambaréné (CERMEL), Lambaréné BP 242, Gabon; aadegnika@gmail.com (A.A.A.); m.p.grobusch@amsterdamumc.nl (M.P.G.); 2Center of Tropical Medicine and Travel Medicine, Department of Infectious Diseases, Amsterdam University Medical Centers, Location AMC, Amsterdam Infection &Immunity, Amsterdam Public Health, University of Amsterdam, P.O. Box 22660, 1100 DD Amsterdam, The Netherlands; 3Institut für Tropenmedizin, German Center for Infection Research (DZIF), Eberhard Karls Universität Tübingen, 72074 Tübingen, Germany; 4African Institute for Development Policy, Lilongwe, Malawi; paul.kawale@gmail.com (P.K.); levi.kalitsilo@afidep.org (L.K.); 5Department of Clinical Sciences, Liverpool School of Tropical Medicine, Liverpool L3 5QA, UK; shevin.jacob@lstmed.ac.uk (S.T.J.); jamie.rylance@lstmed.ac.uk (J.R.); 6Walimu, Kampala, Uganda; 7Malawi-Liverpool-Wellcome Trust, Blantyre, Malawi; 8Department of Parasitology, Leiden University Medical Center, 2333 ZA Leiden, The Netherlands; 9Masanga Medical Research Unit, Masanga, Sierra Leone; 10Institute of Infectious Diseases and Molecular Medicine, University of Cape Town, Cape Town 7925, South Africa

**Keywords:** sepsis, policy engagement, prioritisation, health system, Lambaréné, Gabon

## Abstract

Sepsis has been recognised as a global health priority by the United Nations World Health Assembly, which adopted a resolution in 2017 to improve sepsis prevention, diagnosis, and management globally. This study investigated how sepsis is prioritised in Gabon. From May to November 2021, we conducted a qualitative study in healthcare stakeholders at the local, regional, and national levels. Stakeholders included the Ministry of Health (MOH), ethics/regulatory bodies, research institutions, academic institutions, referral hospitals, international funders, and the media. Twenty-three multisectoral stakeholders were interviewed. Respondents indicated that sepsis is not yet prioritised in Gabon due to the lack of evidence of its burden. They also suggest that the researchers should focus on linkages between sepsis and the countries’ existing health sector priorities to accelerate sepsis prioritisation in health policy. Stakeholder awareness and engagement might be accelerated by involving the media in the generation of communication strategies around sepsis awareness and prioritisation. There is a need for local, regional and national evidence to be generated by researchers and taken up by policymakers, focusing on linkages between sepsis and a country’s existing health sector priorities. The MOH should set sepsis reporting structures and develop appropriate sepsis guidelines for identification, management, and prevention.

## 1. Introduction

Sepsis is a life-threatening organ dysfunction resulting from infection [1]. Sepsis kills 11 million people each year [2]. The term “sepsis” dates back to Hippocrates’ time, when it was used to describe the process through which flesh rots and wounds fester [3]. Despite this long history, countless patients around the world continue to die of sepsis or suffer long-term disability [4,5]. The World Health Organization (WHO) recommends strengthening efforts to identify, document, prevent, and treat sepsis [6].

Recently, sepsis care has substantially improved due to extensive research efforts allowing for novel insights into the pathophysiology, treatment, and awareness of sepsis [7,8,9,10,11,12,13]. In 2017, a WHO resolution recommended that member states recognise sepsis as a Global Health Priority [6,14]. The resolution also encourages health workers to increase sepsis awareness by using the term ‘sepsis’ in communication with patients, relatives, and other parties [6,14]. Many barriers delay the reduction in the global burden of sepsis, particularly in low-resource settings [15,16]. Many studies consistently report low community and stakeholder awareness of sepsis, its signs and symptoms, its causes, and resulting disability and death toll [17,18,19,20]. To be engaged, health leaders, researchers, and funding agencies need accurate quantification of sepsis incidence and mortality. We hypothesised that prioritising sepsis in Low-and-Middle-Income Countries (LMICs) such as Gabon might face many challenges or reticence from stakeholders. Using a qualitative approach, we collected Gabon’s health system stakeholders’ opinions on prioritising sepsis and used it as an indicator of the health system’s performance. Furthermore, we aimed to elucidate baseline perceptions of key health workers on the burden of sepsis in Gabon and to identify opportunities for stakeholder engagement.

## 2. Materials and Methods

### 2.1. Study Design and Study Site

From May to November 2021, we conducted a qualitative study at the Centre de Recherches Médicales de Lambaréné (CERMEL), Gabon. Participants included stakeholders at the local, regional, and national levels. They were from Ministry of Health (MOH), ethics/regulatory bodies, research institutions, academic institutions, referral hospitals, international funders, and media. These stakeholder institutions have been selected based on their reported role in health policy in general, and on their contribution to improving and developing solution to health challenge at both local regional and national levels [21,22,23,24]. This study is part of the policy engagement component of the African Research Collaboration on Sepsis (ARCS), a multinational research initiative funded through the UK National Institute for Health Research (NIHR), which aims to: (1) deliver high-quality sepsis research training; (2) establish commonly agreed sepsis care quality indicators for Gabon, which could form the bedrock of monitoring and evaluation programmes; (3) and pilot test innovative sepsis care interventions [25].

### 2.2. Participant Selection

Letters were sent to institutions identified as employing members of stakeholder groups representing the health sector, requesting that they suggest names of key informants who met the following eligibility criteria: being 18 years of age or more; having ever heard of sepsis; being willing to provide informed consent for the interview. These criteria were required to make sure that participants could provide their opinions based on their experience in their respective institutions.

### 2.3. Data Collection

A qualitative semi-structured questionnaire was used in face-to-face key informant interviews. The questionnaire was adapted from the sample Bellwether tool [26] (Appendix A). Study participants were asked for: their perceptions of policy agenda priorities; characteristics and capacities of sepsis-related policymakers, policy implementers, advocates, opponents; and factors that might elevate sepsis on the policy agenda. These key informant interviews were audio-recorded, annotated, and transcribed. Study participants also completed a quantitative tool, which asked them to rate, for each stakeholder group, the likelihood of sepsis-related outcomes being realised in the next five years, on a scale of 1–5 (where 1 = highly unlikely; 2 = unlikely; 3 = neither likely or unlikely; 4 = likely, and 5 = highly likely). They took into consideration what influential MOH policymakers are saying about sepsis, what language they are using, how interested and open MOH policymakers are to sepsis, and what kind of evidence would convince them. They also took into consideration who, other than the MOH, is engaging in sepsis and how influential they are; what can be done to involve others; what influential stakeholders are saying about sepsis; and what new legislation, budgets, programmes, or strategies were being developed that could relate to sepsis. Regarding health workers specifically, participants considered who is involved in implementing sepsis-related policies among health workers; whether they have the skills, relationships, and incentives to deliver; whether different health workers are working coherently together to implement sepsis-related policy; and whether the necessary structures and incentives are in place to facilitate this (Appendix A).

### 2.4. Data Analysis

Framework Analysis was employed for the qualitative data collected from the key informant interviews. Analysis of the data was structured into five phases: familiarisation, identifying a thematic framework, indexing, mapping, and interpretation. During the thematic framework identification phase, we used the interview guide in a deductive process of identifying broad themes. This thematic framework was refined inductively by identifying emerging themes. NVivo software [27] was used to summarise emerging themes.

## 3. Results

### 3.1. Characteristics of Included Participants

Twenty-three stakeholders from seven institutions were interviewed. A total of 11/23 (48%) of the participants were aged between 31 and 40 years old, 16/23 (70%) were males, and 10/23 (43%) and 13/23 (57%) held Bachelor and Doctorate degrees, respectively. The distribution of stakeholder by interviewed institution is presented in Table 1.

### 3.2. Health Priority in the Gabon Ministry of Health’s Agenda

We asked participants to identify the top three priorities for the Ministry of Health. Figure 1 shows the word cloud of participants’ responses. Most respondents mentioned health promotion priorities for the prevention, diagnosis, and surveillance of infectious diseases (COVID-19, HIV, antimicrobial stewardship, tuberculosis, malaria); maternal and child health priorities; and chronic non-communicable diseases. None of the participants mentioned sepsis as a priority health issue. Malaria, tuberculosis, and COVID-19 are the three most-frequently mentioned diseases with 30% (7/23), 22% (5/23), and 13% (3/23) proportions, respectively.

### 3.3. Gabon Sepsis Policy Strengthening

Study participants also estimated the likelihood of sepsis-related policy outcomes for stakeholder groups, as shown in Table 2. The participants responded that it is highly likely that the MOH will demand: evidence on sepsis; agreement by training institutions on a definition of sepsis; participation by health workers from central hospitals and health worker unions in sepsis training; permission granted by ethics committees for researchers to audit patients’ clinical records; and resultant sepsis evidence supplied by researchers. Conversely, study participants perceived that the least likely outcome is that the MOH and regulatory bodies will recognise sepsis as a priority disease and put sepsis as an indicator of the quality of the health system.

The results from the qualitative interviews were further presented using three ‘building blocks’ of WHO health system strengthening [28], namely, service delivery, leadership, and governance, finance, and information.

#### 3.3.1. Service Delivery

Respondents in our study indicated that referral hospitals lost many sepsis patients because there are no clear guidelines and indicators for reporting the various types of sepsis. Physicians and nurses use clinical judgment for diagnosis. Some participants acknowledged that they were aware of the recent sepsis score during the sepsis project performed in their hospital by CERMEL’s research team. Training institution stakeholders estimated that it is highly likely (rated 5) for training institutions to agree on a definition of sepsis and teach how to accurately diagnose and report sepsis. As a result, most of the respondents from the health workers’ group estimated that it is likely (rated 4) for them to accurately diagnose and report sepsis.


*“I think it is important to train people, including the nurses, in the diagnosis of sepsis because it is not well known, as soon as someone has a fever, we go after malaria, we forget everything else. Sometimes, the thick blood smear is negative, but people are treated for malaria, so I think there is a problem with the training. And for those who are trained, knowledge updates are needed from time to time.”*
—Physician from research institution.

#### 3.3.2. Leadership and Governance

Respondents were not aware of a sepsis policy that includes prevention, treatment, and rehabilitation. They recommended advocating for the MOH to lead the development of national policy and guidelines. Most respondents indicated that it is unlikely (rated 1) for MOH to put sepsis care as an indicator of the quality of Gabon’s health system. However, most of our respondents estimated the likelihood of state regulatory bodies to recognise sepsis as a priority disease to be highly likely (rated 5).


*“As I said, it is first of all the evidence. Really to highlight the impact of sepsis in the management of child mortality. Once this is automatically demonstrated, these figures will push the ministry to put in place strategies to combat sepsis. And once the ministry has registered action as a priority in its program, there is no need to go to the National Assembly to set it. For now, we are not there. Sepsis could be set as indicator for infection control program but not the whole health system.”*
—Medical microbiologist from MOH.

Most of the respondents also estimated that it was likely (rated 4) that research institutions will agree to conduct more multi-disciplinary clinical research on sepsis to address the lack of evidence of sepsis burden.

#### 3.3.3. Finance

According to our respondents, funders might be interested in sepsis if there is evidence of the burden in the population. They estimated that funders are likely (rated 4) to convene stakeholders’ meetings on sepsis.


*“The first thing I think is to generate the data. From my point of view, the data exist since the hospital structures are confronted with this problem. It is therefore necessary to do this collection work at the national level in order to measure the extent of the problem. We must involve training and research institutions because there are two elements in Africa, the absence of data but also the support, and for this point, it is necessary that the institutions know that there is this problem in order to focus from initial and continuing training and research institutions to generate data. All this comes at a cost–therefore, the Ministry of Finance and Budget, the insurance companies that pay for these diseases, must be involved. Once these pre-requisites have been established, the funder will be able to financially support the Country.”*
—Participant from international funder institution.

#### 3.3.4. Information

The media participants estimated the likelihood of disseminating evidence on sepsis, implementing public engagement activities on sepsis, and commemorating World Sepsis Day at highly likely (rated 5). However, a lack of awareness of sepsis was noted by the public and media, as explained by this journalist:


*“As a journalist, I do not have much knowledge in the field of sepsis. It is a bit difficult to assist you in improvement of the community awareness on sepsis without a minimum knowledge from my side. You have to think about involving the journalists by improving our knowledge on sepsis.”*
—Journalist at a media institution.

Our respondents also cited patients, survivors of sepsis, as the most appropriate advocates. They could inform the community on sepsis and improve awareness.


*“*
*In my opinion, if the thing is well explained to the population and if the population is well exposed to this condition, I think that people will buy in and maybe lift any reservations they may have. If I can rely on the experience of COVID-19, many people did not believe in COVID-19 because they did not see sick people. Basically, we said, we don’t see anyone sick with COVID-19. We do not see COVID-19 deaths so COVID-19 does not exist. One of the ways to overcome this reluctance of the populations would be to demonstrate to the populations that the disease does exist, that there are patients who exist, that there are people who suffer from it and also that there are a people who can die of it. That is to say, make it much more tangible or concrete at the level of the population.”*
—Journalist at a media institution.

### 3.4. Prioritising Sepsis in Gabon’s Health System

On considering sepsis as a health priority, all but one participant agreed. The exceptional stakeholder thought that there is not enough evidence for such a conclusion.


*“Considering the number of cases I have been confronted with, I would say no. But I might not be aware about what happens with other hospitals. With regard to what the various health services declare sepsis might be a real problem but not a priority.”*
—Public health specialist from international funder institution.

The respondents also identified the categories of people or groups as main advocates or opponents for sepsis being set as a health priority. Suggested advocates included medical doctors, nurses, paediatricians, infectious diseases specialists, microbiologists, intensive care specialists, non-governmental organisation representatives, and sepsis survivors.

Despite all participants reporting that it is highly likely that sepsis could be considered a health priority in Gabon, they were quick to state that there is a need for strong evidence/data to convince the MOH.

## 4. Discussion

We collected stakeholder opinions on prioritising sepsis in the Gabon health system. Our study has found that sepsis is perceived to not be a priority among most health system stakeholders. Although the WHO has declared that sepsis is a global health problem, there is variation in critical care resource allocation for the management of sepsis in LMICs [29,30]. This indicates that regions and countries do not necessarily follow the same priorities. As reported previously by other studies, the absence of a harmonised definition contributes to delayed prioritisation in many LMICs’ health systems [31,32]. The situation is aggravated by the lack of local or country-level evidence to justify the adoption of sepsis as a priority.

Our study shows that most of the stakeholders perceived it to be highly likely that sepsis could be considered as a health priority by Gabon’s MOH if the evidence of its burden as a public health issue is demonstrated. The Global Sepsis Alliance and African Sepsis Alliance have advocated for high-quality evidence on sepsis in Africa. In the coming years, robust data on sepsis from many African countries will be published [33,34,35]. The research center in Gabon CERMEL have provided local pieces of evidence on the burden of sepsis [36,37,38,39,40,41,42]; however, there is a need for national-level evidence. These data could serve as a baseline to guide public policy. The lack of evidence in LMICs affects all levels of healthcare delivery from individual patient management to strategic planning at the health-system level [19].

Stakeholders in our study estimated that sepsis would be unlikely to be set as a quality indicator of the health system in Gabon. While there is no single indicator of health system performance, sepsis is a common and final pathway of many infections causing death, including COVID-19. It could help to have a good overview of infectious disease prevention strategies, including hand hygiene, immunisation programme, chemoprophylaxis, food safety, safe water and sanitation, injection safety and sterilisation, blood safety, and vector control [43,44].

One of the major findings of our study was that health workers and stakeholders strongly advise involving the community, non-governmental organization representatives, sepsis survivors, and the media in policy engagement activities. It is clear that media participants were relatively unaware of sepsis as a public health issue. Interestingly, the three most-mentioned diseases (malaria, tuberculosis, COVID-19) as health priorities are those well mediated by the health authority and more prevalent in Gabon. Opportunities and vehicles for raising awareness such as World Sepsis Day could be used to engage the media, which would then feed to wider stakeholder awareness.

The strengths of our study were that we interviewed representative stakeholders from all levels of the Gabon health system, and that the interviews were all carried out by the same interviewer, ensuring consistent questioning, but the open question format also allowed for insight into the actual awareness.

However, due to the face-to-face interview method, some participants might have been reluctant to answer questions openly. We did not perform a focus group discussion to further appreciate the opinions of stakeholders. These limitations would not significantly affect our findings. To the best of our knowledge, this study is the first in Gabon and the Central Africa region to assess the stakeholder opinion on prioritising sepsis in the health system. Our study calls for performing nationwide epidemiological quantitative studies to investigate the burden of sepsis in Gabon, the cost-effectiveness of considering sepsis as a health priority, and the role of the community in preventing sepsis or reducing sepsis mortality. Beyond the national level, our study shows the need for international advocacy for more funding for research on sepsis in LMICs, for adding sepsis in WHO Global Burden of Disease Report.

## 5. Conclusions and Recommendations

Despite calls across the international public health community for positioning sepsis as a global health priority, our study illustrates that sepsis is not yet within the health priorities of Gabon. There is a need for local, regional and national evidence to be generated by researchers and taken up by policymakers, focusing on linkages between sepsis and a country’s existing health sector priorities. We recommend that the MOH uses this evidence to set sepsis indicators and reporting structures and develop appropriate sepsis guidelines for identification, management, and prevention. The media should be involved for the generation of a communications strategy that will contribute to the acceleration of sepsis awareness by key stakeholders. Future perspectives include conducting cost-effectiveness studies for setting sepsis as a global health priority, as well as implementation studies for improving of stakeholder awareness and routine hospital-based sepsis data reporting.

## Figures and Tables

**Figure 1 healthcare-10-00877-f001:**
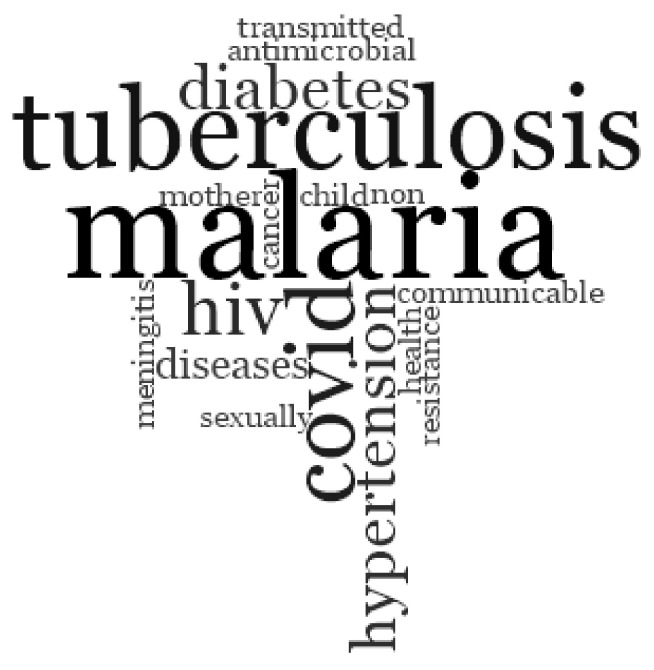
Perceptions of stakeholders on disease prioritisation in Gabon. The size of the writing of the disease is proportional to the number of participants who mentioned it as health priority. Malaria, tuberculosis and COVID-19 were most frequently mentioned.

**Table 1 healthcare-10-00877-t001:** Distribution of stakeholders interviewed according to institution.

Employing Body	Physicians	Nurses	Public Health Specialist	Laboratory Technician	Journalist	Economist	Total
**Stakeholder Group**	8	3	4	4	3	1	23
Ministry of Health	1		1	1			3
Ethics/Regulatory Body		1	1	1			3
Research Institution	3						3
Training Institutions	2	2					4
Referral Hospital	1		1	2			4
International funder	1		1			1	3
Media					3		3

**Table 2 healthcare-10-00877-t002:** Participants’ estimation of the likelihood of intervention to be pursued to elevate sepsis to a health priority in Gabon (*N* = 23).

Stakeholder Group (Number of Participants Interviewed)	Outcome	Likelihood of Intervention (Mode)		
		Unlikely	Neutral	Highly Likely
MOH (3)	Demand evidence on sepsis	0	0	3
	Organise training workshops on sepsis for health workers	0	0	3
	Put sepsis as the indicator of quality of health system	3	0	0
Referral hospitals (4)	Participate in training workshops on sepsis	0	0	4
	Accurately diagnose and report sepsis	0	0	4
	Recognition of sepsis as a priority disease	0	0	4
Ethics committees (3)	Give permissions for clinical audit of patient records for quality improvement	0	0	3
	More multi-disciplinary clinical research on sepsis	0	0	3
	Recognition of sepsis as a priority disease	1	0	2
Research organisations (3)	Supply evidence on sepsis	0	0	3
	Implement policy engagement activities on sepsis	0	0	3
	Organise conference tracks on sepsis	0	0	3
Training institutions (4)	Agree on a definition of sepsis	0	0	4
	Teach how to accurately diagnose and report sepsis	0	0	4
	Future health workers recognise sepsis as a priority disease	0	0	4
International funders (3)	Convene stakeholders’ meetings on sepsis	0	0	3
	Put out calls for proposals around sepsis	1	0	2
	Put sepsis on international donors’ agenda	0	0	3
Media (3)	Disseminate evidence on sepsis	0	0	3
	Implement public engagement activities on sepsis	0	0	3
	Commemorate World Sepsis Day	0	0	3

MOH: Ministry of Health.

## Data Availability

The data presented in this study are available from the corresponding author upon request.

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
