# Peer review of "Recognising Sepsis as a Health Priority in Sub-Saharan African Country: Learning Lessons from Engagement with Gabon’s Health Policy Stakeholders"

_healthcare, 2022, doi:10.3390/healthcare10050877_

Round 1

Reviewer 1 Report

  1. Abbreviations have to be provided in full name the first time they are used with the short text in brackets. Please check and apply across the document
  2. In this research, the reason of participants selection eligibility criteria need to be specified with proper citation or valuable literatures
  3. Detailed discussion of figure 1 required to be improved
  4. In the results section, critical analysis of results need to be mentioned for future direction
  5. In Conclusion and Recommendations, authors can add 1-2 good future directions

Author Response

We thank the reviewer for commenting on our manuscript intituled “Recognising Sepsis as a Health Priority in Sub-Saharan African Country: Learning Lessons from Engagement with Gabon’s Health Policy Stakeholders” (healthcare-1646949). We were happy to receive valuable comments which are improving the quality of the manuscript significantly.

Abbreviations have to be provided in full name the first time they are used with the short text in brackets. Please check and apply across the document

Reply: We thank the reviewer for the comment. The abbreviations have been defined accordingly.

In this research, the reason of participants' selection eligibility criteria needs to be specified with a proper citation or valuable literature

Reply: Thank you for this valuable comment. We have done so and added the following sentence to the study design and participant selection section:  

“These stakeholder institutions have been selected based on their reported role in health policy in general, and on their contribution to improving and developing solutions to health challenges at both local and national levels [21–24]”.

“These criteria were required to make sure that participants could provide their opinions based on their experience in their respective institutions.”

Detailed discussion of figure 1 required to be improved

Reply: We are now providing more detail in the legend and expanding the discussion of the respective data.

In the results section, a critical analysis of results needs to be mentioned for the future direction

Reply: We thank the reviewer for the comment. The following sentences have been added to the discussion:

“One of the major findings of our study was that health workers and stakeholders strongly advise involving the community, non-governmental organisations, sepsis survivors, and media in policy engagement activities. It is clear that media participants were relatively unaware of sepsis as a public health issue. Interestingly, the three most mentioned diseases (malaria, tuberculosis, COVID-19) as health priority are those well mediated by the health authority and more prevalent in Gabon.”

“Our study calls for performing nationwide epidemiological quantitative studies, to investigate the burden of sepsis in Gabon, the cost-effectiveness of considering sepsis as a health priority, and the role of the community in preventing sepsis or reducing sepsis mortality. Beyond the national level, our study shows the need for international advocacy for more funding for research on sepsis in LMICs, for adding sepsis in WHO Global Burden of Disease Report.”

In Conclusion and Recommendations, authors can add 1-2 good future directions

Reply: The following sentences have been added to the discussion:

 “Future perspectives include conducting cost-effectiveness studies for setting sepsis as a global health priority, as well as implementation studies for improving stakeholder awareness and routine hospital-based sepsis data reporting.”

Reviewer 2 Report

Page 2--line 3--a WHOresolution recommended that member states recognise sepsis.....

           2.3 Data Collection--line 5----.. advocates, opponents, and factors that might elevate sepsis....

page 3-- 2.4 Data Analysis...line 6---.....summarise

page 5-line 13----Most of the respondents. also estimated that it is likely (rated 4) that research institutions.....

Author Response

We thank our reviewer. All typos have been corrected accordingly and the manuscript has been revised in detail, weeding out further typos.

Round 2

Reviewer 1 Report

The reviewer's comments have been corrected by the author.